# Exploring the Link between Social Support and Patient-Reported Outcomes in Chronic Obstructive Pulmonary Disease Patients: A Cross-Sectional Study in Primary Care

**DOI:** 10.3390/healthcare12050544

**Published:** 2024-02-25

**Authors:** Izolde Bouloukaki, Antonios Christodoulakis, Katerina Margetaki, Antonia Aravantinou Karlatou, Ioanna Tsiligianni

**Affiliations:** 1Department of Social Medicine, School of Medicine, University of Crete, 71003 Heraklion, Greece; christodoulakisa@icloud.com (A.C.); katmargetaki@hotmail.com (K.M.); toniakoinerg2@gmail.com (A.A.K.); i.tsiligianni@uoc.gr (I.T.); 2Department of Nursing, School of Health Sciences, Hellenic Mediterranean University, 71410 Heraklion, Greece

**Keywords:** social support, PROMs, COPD, primary care

## Abstract

We aimed to explore the link between social support and various patient-reported outcome measures (PROMs) in primary care patients with COPD. This was a cross-sectional study with 168 patients with COPD from six primary care centers in Crete, Greece. We collected data on sociodemographic characteristics, medical history, disease-specific quality of life, the COPD Assessment Test (CAT), fatigue, the Fatigue Severity Scale (FSS), phycological parameters, Patient Health Questionnaire-9, General Anxiety Disorder-7, sleep complaints, the Pittsburg Sleep Quality Index, the Athens Insomnia scale (AIS), and the Epworth Sleepiness Scale. Social support was measured using the Multidimensional Scale of Perceived Social Support (MSPSS). Out of 168 patients with COPD, 114 (68.9%) exhibited low levels of social support. Low social support (MSPSS total ≤ 5) was positively associated with COPD symptoms (CAT score ≥ 10) (OR = 3.97, 95%CI:1.86–8.44; *p* < 0.01), fatigue (FSS ≥ 36) (OR = 2.74, 95%CI:1.31–5.74; *p* = 0.01), and insomnia symptoms (AIS ≥ 6) (OR = 5.17 95%CI:2.23–12.01; *p* < 0.01), while the association with depressive symptoms (PHQ-9 ≥ 10) was marginally significant (OR = 3.1, 95%CI:0.93–10.36; *p* = 0.07). Our results suggest that lower levels of social support are positively associated with PROMs in patients with COPD. Therefore, our findings show an additional way to improve the overall health of patients with COPD in primary care by putting social support at the epicenter of actions.

## 1. Introduction

Chronic obstructive pulmonary disease (COPD) is a highly prevalent and progressive respiratory disease, resulting in substantial morbidity and mortality [1,2,3]. The worldwide prevalence of COPD is approximately 10.3%, but this number may increase as smoking becomes more common in low- and middle-income countries and as the population ages in high-income countries [4]. In Greece, it has been approximated that around 8.4–10.6% of the Greek population is affected by COPD, with a particularly pronounced impact on the elderly and those residing in rural regions [5,6]. This is important, since COPD is characterized by severe symptoms such as coughing, breathlessness, increased sputum production, and reduced physical activity [7,8]. Additionally, symptoms such as persistent fatigue, unintentional weight loss, and disrupted sleep patterns, including insomnia, can further exacerbate their emotional well-being [9,10,11].

Sleep disturbances are a relatively common symptom of COPD that can have detrimental effects on the quality of life of the patients [12]. For example, chronic insomnia is frequently reported, with up to half of the patients indicating difficulties in initiating sleep, maintaining sleep, or achieving restorative sleep [13]. The significant impact of these sleep disturbances further increases the already heightened risk of anxiety and depression, doubling the likelihood of experiencing them [14]. In addition, the combination of depression and sleep disturbances in patients with COPD leads to a five-fold increase in the likelihood of hospitalization [14]. All these symptoms significantly affect the emotional well-being of patients and elicit feelings of anxiety and depression that ultimately lead to a sense of social isolation [11]. Despite the aforementioned, an assessment of sleep disturbances is not a standard practice in patients with COPD [14].

Healthcare professionals assess the symptoms of patients with COPD by utilizing patient-reported outcome measures (PROMs) during clinical practice [15,16,17,18,19]. These standardized questionnaires provide valuable insights into patients’ perceptions of their health and disease [20]. Furthermore, by utilizing PROMs, healthcare professionals can gain a better understanding of a patient’s health status and where to intervene when necessary [18]. This understanding allows healthcare professionals to address the multiple aspects of COPD, such as physical, emotional, and social, and thus deliver better quality care to their patients. However, despite the fact that PROMs are highly valuable, they do not fully capture the important aspects of COPD, such as social support [21], which seems to be positively associated with mental health, quality of life, and self-efficacy in these patients [17].

The concept of social support refers to the expectation that one’s social network is readily available to offer both emotional and practical assistance when needed [22]. Additionally, it involves providing emotional, informational, and practical assistance to individuals through social networks, including family, friends, peers, healthcare professionals, and community organizations [23]. The available evidence indicates that social support has the potential to enhance stress management and self-esteem, foster medication adherence [22,24], and affect and improve the efficacy of therapeutic interventions [25], quality of life [26], and the well-being of patients with COPD [27]. Furthermore, social support seems to improve the patients’ physical health, alleviate symptoms of depression and anxiety, and reduce hospitalization and mortality rates [28,29]. This could mean that social support has the potential to influence PROMs, given its association with all these aspects and symptoms of COPD, including the sleep disturbances [21,30,31]. Considering the potential influence of adequate social support on patients with COPD [32], it is logical to assume that higher levels of perceived social support could positively impact a range of PROMs for these patients [32]. Nevertheless, there is a lack of comprehensive studies in the literature that sufficiently explore the association between perceived social support and self-reported health in patients with COPD using PROMs, particularly in primary care settings [33]. Therefore, our study aimed to explore the link between social support and various PROMs among patients with COPD in primary care settings.

## 2. Materials and Methods

### 2.1. Design and Sample

The present cross-sectional study invited patients with COPD from six primary care centers in Crete, Greece. To be eligible for inclusion, patients were required to meet the following criteria: (a) be 40 years of age or older and have a physician-diagnosed COPD confirmed with spirometry, (b) have an educational background beyond elementary school, and (c) provide written informed consent. We excluded patients who had severe neurological or mental disorders, were pregnant, experienced a recent exacerbation of COPD, demonstrated limited comprehension of the Greek language, or did not wish to participate.

### 2.2. Procedure

During their visit, patients with COPD were provided with information regarding the objectives of the study. Following their agreement to participate, they proceeded to submit written consent and anonymously completed the questionnaires. To mitigate the influence of social desirability bias, the participants were instructed to deposit their study materials in an opaque container that was placed outside the office.

The study adhered to the guidelines specified in the Declaration of Helsinki and received approval from the University of Crete Research Ethics Committee (REC-UOC) (Protocol Number: 183/13.12.2022).

### 2.3. Data Collection

A comprehensive evaluation of the participants was conducted, which assessed various demographic parameters, such as age, gender, BMI, exercise habits, tobacco and alcohol use, educational status, and comorbidities. COPD-specific quality of life was assessed using the COPD Assessment Test (CAT). The assessment of fatigue involved the use of the Fatigue Severity Scale, while psychological factors were measured using the Patient Health Questionnaire-9 and General Anxiety Disorder-7 questionnaire. Subjective sleep quality and sleep-related complaints were evaluated using the Pittsburgh Sleep Quality Index, the Athens Insomnia Scale, and the Epworth Sleepiness Scale. The multidimensional scale of perceived social support was used to quantify social support. The evaluation for each participant lasted approximately 20–30 min.

### 2.4. Study Tools and Outcomes

#### 2.4.1. Multidimensional Scale of Perceived Social Support (MSPSS)

The MSPSS is a widely used tool designed to assess an individual’s perception of the availability of social support [34]. This scale measures the extent of support received from three specific sources: family, friends, and a significant other. It is comprised of twelve items and three subscales. The total mean scores range from 1 to 7. Higher scores indicate a higher level of perceived social support. The present study employed the total mean score and a cutoff point of 5 or below to determine the presence of low social support [35]. The MSPSS exhibits a high level of internal consistency reliability, as indicated by a Cronbach’s alpha of 0.85–0.91 [36]. The questionnaire has been translated and culturally adapted into Greek [37], and our study has achieved excellent internal consistency with a Cronbach’s alpha of 0.96.

#### 2.4.2. COPD Assessment Test (CAT) Questionnaire

The CAT is a simple-to-complete questionnaire that assesses the self-reported impact of COPD on health status. The CAT consists of eight items (cough, phlegm, chest tightness, breathlessness, limited activities, confidence in leaving home, sleeplessness, and energy), that the patient rates on a scale of 0 to 5 [16]. The score ranges from 0 to 40, with higher values indicating poorer health status. A cutoff point of 10 or above is used to determine the presence of poor health status. It has been also validated in Greek [38] and exhibited a Cronbach’s alpha of 0.78 in our study population.

#### 2.4.3. Patient Health Questionnaire (PHQ-9)

PHQ-9 is a self-report questionnaire that consists of nine items. These items reflect the criteria used in diagnosing depressive disorder according to the Diagnostic and Statistical Manual of Mental Disorders (DSM) 4th edition (DSM-IV) [39] but are theoretically in line with the 5th version of the DSM (DSM-5) [40]. The PHQ-9 has a score range of 0 to 27, with higher scores indicating more severe depressive symptoms. The cutoffs of 5, 10, 15, and 20 correspond to mild, moderate, moderately severe, and severe depressive disorders, respectively. The present study employed a cutoff point of 10 or above to determine the presence of depressive symptoms. The initial validation study of the PHQ-9 showed high reliability in a large sample of primary care patients, with a Cronbach’s alpha coefficient of 0.89 [39]. Subsequent studies in various populations have demonstrated an adequate internal consistency (α = 0.70–0.93) [41,42,43,44,45,46]. It has also been validated in Greek [47] and exhibited a Cronbach’s alpha of 0.86 in our study population.

#### 2.4.4. General Anxiety Disorder (GAD-7)

The GAD-7 is a validated instrument that evaluates seven symptoms related to generalized anxiety disorder, as described in the DSM-IV [48]. The total score ranges from 0 to 21, with higher values indicating greater disturbance. Scores of 5, 10, 15, or higher represent mild, moderate, or severe impairment, respectively. In our study, we considered a threshold above 10 to be a marker for the presence of GAD, indicating the existence of anxiety symptoms at moderate and severe levels. The GAD-7 has been found to have good-to-excellent reliability in different populations, with Cronbach’s α values ranging from 0.8 to 0.97 [49,50,51,52,53,54,55]. The Greek version of the GAD-7 [56] was employed in the current study and yielded a Cronbach’s alpha of 0.93.

#### 2.4.5. Pittsburgh Sleep Quality Index (PSQI)

The PSQI is a widely recognized questionnaire for its ability to distinguish between poor and good sleep quality. The PSQI is a nineteen-item self-rated questionnaire used to evaluate subjective sleep quality and quantity, sleep habits associated with quality, and the occurrence of sleep disturbances among adults within a one-month interval [57]. The score range is 0 to 21. The higher the score, the more pronounced the adverse effects on the sleep quality. A global score of ≥6 indicates poor sleep quality. Additionally, it has been translated and validated in Greek and used for assessing sleep quality in a Greek sample [58,59]. Previous studies have demonstrated satisfactory levels of internal consistency, with Cronbach’s alphas ranging from 0.70 to 0.83 [59,60,61,62,63]. Our study population achieved an acceptable [64] Cronbach’s alpha of 0.67.

#### 2.4.6. Athens Insomnia Scale (AIS)

The AIS is a Greek questionnaire and was designed as a standardized assessment tool to measure the level of sleep difficulty, following the guidelines of the International Classification of Diseases-10 edition (ICD-10). It is a self-assessment psychometric tool comprising eight items. The total score ranges from 0 to 24, and a score equal to or greater than 6 indicates the likelihood of insomnia [65]. In both clinical and community samples, the AIS exhibited exceptional internal consistency, with Cronbach’s alphas ranging from 0.81 to 0.86, thereby confirming its efficacy in measuring insomnia symptoms [66]. For our population, Cronbach’s alpha was 0.85.

#### 2.4.7. Epworth Sleepiness Scale (ESS)

The ESS is presently the most extensively employed subjective assessment tool for measuring daytime sleepiness in clinical settings [67]. The scale ranges from 0 to 24, and anything below 10 is considered to be within the normal range. The Cronbach’s alpha values of previously published studies on ESS were analyzed using reliability generalization meta-analysis, and the cumulative Cronbach’s α ranged from 0.80 to 0.83, indicating a high level of reliability. In our study, the Greek version of the ESS was utilized [68], yielding a Cronbach’s α of 0.75 for this particular population.

#### 2.4.8. Fatigue Severity Scale (FSS)

The FSS is used by providing individuals with nine statements concerning the severity, frequency, and impact of fatigue on daily life (physical functioning, exercise and work, and family or social life) and asking them to rate their agreement. The FSS score is obtained by finding the average of the scores given to each item, with higher scores reflecting more severe fatigue. Scores < 36 were considered normal. A score that surpasses the specified threshold (≥36) indicates a substantial detrimental impact of fatigue on daily life activities (maximum score of 63) [69]. The FSS has been translated and culturally adapted to Greek [70]. Furthermore, the FSS has been validated in various populations, and the findings suggest that it has satisfactory concurrent validity and internal consistency, with Cronbach’s alphas ranging from 0.89 to 0.96. The Cronbach’s alpha for our population was 0.99.

### 2.5. Statistical Analysis

Participants who had completed the MSPSS questionnaire (N = 168) were included in our analysis. For all continuous variables with a normal distribution, the results are reported as the mean ± standard deviation (SD), whereas for variables without a normal distribution, the median (25–75th percentile) is presented. Categorical variables are presented in terms of the absolute numerical value and the corresponding percentage. To conduct comparisons between groups, we utilized a two-tailed *t*-test for independent samples (when data followed a normal distribution) or a Mann–Whitney U-test (when data did not follow a normal distribution) for continuous variables. Furthermore, the Chi-square test was employed for categorical variables. Continuous scales were correlated using the Pearson’s correlation coefficient.

In order to assess the associations between the MSPSS and the studied PROMs, we employed linear regression for the continuous MSPSS scales and logistic regression for the dichotomized MSPSS scales. Each PROM was then fitted into a separate model. In each model, we also included factors that were associated (*p* < 0.05) with both the MSPSS and PROMs. Thus, all models were adjusted for age, gender, marital status, and level of education. We further adjusted for obesity and the presence of any other chronic diseases in a sensitivity analysis. Multicollinearity among the predictors was assessed using collinearity statistics to ensure that the collinearity between the predictor variables was within the acceptable range, as indicated by the tolerance value variance inflation factor (VIF < 3 for each model). The results were deemed significant if the *p*-values were less than 0.05. Data were analyzed using the Stata software (version 13).

## 3. Results

### 3.1. Patient Characteristics

A total of 191 patients with COPD were initially invited to participate in the study; however, 170 patients (89%) agreed to participate. Moreover, the study questionnaires were completed by 168 patients, yielding an effective response rate of 88%. The average age of the patients included in the study was 68 years (range 41–90 years). Among the participants, 68% were male, 41% were obese (BMI ≥ 30 kg/m2), and 40% were married. A significant portion of the participants exhibited a low level of education (48%), a slightly smaller percentage had a middle education (35%), and only 17% had a high level of education. In terms of smoking habits, 46% of the participants were actively smoking during the survey, whereas 44% had quit smoking. At least one chronic disease was present in 89% of the patients. The sociodemographic characteristics and health status of the 168 participants are provided in Table 1.

Table 2 displays the distribution of patients into the GOLD groups based on the CAT categorization. The majority of participants (52.4%) were classified in the GOLD group B, whereas group E accounted for less than 17%, according to the GOLD 2023 classification. In the 12-month period prior to the study, the majority (85%) of patients did not suffer from COPD exacerbations or had just one, whereas 14.9% experienced two or more exacerbations and 4.8% required hospitalization. The mMRC scores are also presented in Table 2.

Regarding social support, the mean (SD) values of the MSPSS are presented in Table 3. The highest score was observed in the domain of family support, followed closely by the “significant other” domain. The three subscales are highly correlated to each other and to the total social support scale. The highest correlation was observed between the “significant other” subscale and the “family” subscale (rho = 0.887, *p* < 0.001) and the lowest between the “family” and the “friends” subscale (rho = 0.685, *p* < 0.001). Also, the correlations with the total scale were strong and highly significant, ranging from 0.898 for the “friends” subscale to 0.921 for the “significant other subscale” (all *p*-values < 0.001). The “significant other” and the “family” subscales accounted for 86% of the variance in the total social support scale, and this percentage for the “friends” subscale was 80%.

### 3.2. Differences in Clinical Characteristics of Patients with COPD with High and Low Social Support

The application of an MPSS ≤ 5 cutoff to define low social support led to the identification of 32.1% of patients with high social support and 68.9% of patients with low social support. The clinical variables of the two groups are summarized in Table 1 and Table 2. No significant differences were found in the anthropometrics and comorbidities between the groups. However, patients with lower social support had lower educational levels, were slightly older, and had worse COPD health status based on the CAT score and GOLD classification.

### 3.3. Correlation of Social Support with PROMs

With regard to PROMs, it is important to highlight that a substantial proportion of patients displayed elevated levels of fatigue and nighttime symptoms, as assessed using the AIS and PSQI. Furthermore, patients lacking social support exhibited the most severe functional impairments, with statistical significance observed across nearly all the questionnaires (Table 4). It is evident that individuals who lacked adequate social support experienced heightened levels of fatigue, depression, anxiety, symptoms of insomnia, and a poorer quality of sleep compared to those with high support. The ESS score did not differ between the groups. However, the mean ESS score for the entire sample was relatively low, and most of the patients who reported an ESS score > 10 were in the low social support group. Nevertheless, these differences were not statistically significant.

After adjusting for age, gender, marital status, and level of education, it was observed that not only the total score of the MPSSP but also its individual domains displayed an inverse relationship with the CAT, FSS, PHQ-9, GAD-7, and AIS scores (Table 5). Moreover, the presence of low social support was still found to be independently associated with COPD symptoms (CAT score ≥ 10) (OR = 3.97, 95% CI 1.86–8.44; *p* < 0.01), fatigue (FSS ≥ 36) (OR = 2.74, 95% CI 1.31–5.74; *p* = 0.01), and insomnia symptoms (AIS ≥ 6) (OR = 5.17 (2.23, 12.01, 95% CI 2.23–12.01; *p* < 0.01) (Table 6). The association between low social support and the presence of depressive symptoms was close to being statistically significant (OR = 3.1, 95% CI 0.93–10.36; *p* = 0.07), possibly due to the low number of patients with a PHQ-9 score ≥10. The same principle applies to the correlation between symptoms of anxiety and a lack of social support (OR = 2.57, 95% CI 0.82–8.12; *p* = 0.11). Further adjustments for obesity and the presence of any other chronic disease resulted in similar results (Appendix A).

## 4. Discussion

In the present study, we explored the link between social support and various PROMs among patients with COPD in primary care settings. Our findings indicated that a significant proportion of patients with COPD exhibited low levels of social support. Furthermore, lower levels of social support were positively associated with worse health status, increased fatigue, depression, anxiety, and symptoms of insomnia in these patients, independently of participants’ age, sex, obesity, presence of chronic diseases, education, and marital status.

The average level of social support within our population, as measured by MSPSS, was 4.4, indicating a relatively lower level of support compared to the average of 5.1–5.7 among mixed COPD populations including patients with COPD [71,72] or populations with COPD and with comorbid heart failure [73] and/or other diseases [74,75]. On the other hand, our findings were similar (4.2) to a recent study focusing on COPD populations [76]. In our study, the majority of the patients with COPD lacked sufficient social support. This supports previous research showing that among those with COPD, roughly one in six patients underwent social isolation and one in five experienced feelings of loneliness [77]. Fear of being judged and experiencing social stigmatization due to visible symptoms of their disease, such as coughing and the presence of phlegm, are potential contributing factors [78]. Another important point to consider is the findings of a previous study that highlighted that patients with COPD are less likely to have a partner than non-COPD subjects [79]. Furthermore, even among COPD patients who do have a partner, they are less likely to feel “very satisfied” with the daily support provided by their partner [79]. In contrast, our analysis revealed that perceived social support from family members, the domain of family support, received the highest rating score, indicating a higher level of familial support, which is a common trait in interpersonal relationships within Greek society, particularly within Greek families [80]. After all, the Greek family is known for its strong bonds and often serves as the primary source of support and feedback, as well as acting as a protective shield during challenging times [80].

Research has indicated that social support has a significant positive effect on health-related PROMs in individuals with COPD [33]. This was also the case in our study, wherein social support exhibited a notable influence on PROMs encompassing COPD-specific health status, fatigue, mental health, and sleep health. The COPD health status of our population was characterized by a mean CAT score of 12.1, indicating a moderate impact of COPD on health. Additionally, individuals with less social support exhibited a poorer health status related to COPD, as indicated by their higher CAT scores and a greater proportion of individuals falling into GOLD classification groups B and E. These results are in line with previous studies indicating that patients with COPD with high CAT scores (worse health status) had significantly lower scores on social support scales, as evaluated by the Social Provision Scale (SPS) [81] and Social/Family Well-Being domain of the self-reported Functional Assessment of Chronic Illness Therapy-Fatigue questionnaire [82]. This is further corroborated by a recent study conducted in primary care settings, which revealed a positive correlation between CAT scores and low social support [83]. One possible explanation for this finding could be that social support might influence the psychological and physiological mechanisms related to health outcomes [84,85,86,87]. For example, it has been suggested that higher levels of social support in conjunction with the presence of the neuropeptide oxytocin could increase the cortisol levels of patients with COPD in stressful situations, such as exacerbations [84,85,86]. On the other hand, by reducing cortisol levels, these factors may contribute to heightened feelings of calmness and decreased anxiety during stressful situations related to COPD [84,85,86]. Therefore, our results indicate that respiratory-specific quality of life (as measured by the CAT) is positively linked to perceived social support.

Another crucial clinical indicator of health status among patients with COPD is fatigue, which is a complex symptom [88], influencing both physical and mental functioning, quality of life, and perceived control over life [89]. This symptom is highly prevalent, as in a previous study, it was found in nearly half of the patients diagnosed with stable, moderate-to-severe COPD [90], with a range of 17–95% in a recent review [91]. Our study’s participants also exhibited a high prevalence of fatigue, with 65% of participants reporting this symptom. Moreover, individuals who reported symptoms of fatigue were 2.74 times more likely to have low social support. Although the literature does not provide sufficient evidence, previous studies have indicated that patients who have a partner experience less fatigue, whereas widowed individuals tend to experience more severe fatigue than both married and unmarried individuals [92,93]. However, fatigue is linked to more than just social support; it also affects social functioning, leading to social isolation, loneliness, and increased mental burden [89]. This finding could be explained through the social exchange theory [94]. This theory suggests that social interactions can have both positive and negative outcomes, including health-related ones [95,96,97,98]. In this case, patients with COPD could experience some of their symptoms (such as fatigue, depression, and anxiety) more intensely as a result of negative social interactions and possibly a lack of social support [82,96,99]. Therefore, more consideration should be given to social support in future research and clinical practice.

Our study found a relatively lower prevalence of depressive (15.5%) and anxiety symptoms (17.7%), thus highlighting the possibility of mental disorders being overlooked in these individuals. It is well known that individuals with COPD often experience mental health issues such as anxiety and depression, which greatly affect their overall well-being [100,101,102]. It is estimated that approximately 30% of people diagnosed with COPD also experience depression, and between 10% and 50% have coexisting anxiety [100,101]. More importantly, the prevalence of depression (19.3%) and anxiety (21.1%) symptoms was higher in participants with lower social support. Additionally, the higher PHQ-9 and GAD-7 scores (indicating more depressive and anxiety symptoms) showed a significant positive correlation with the presence of low social support. A possible explanation for these findings could be that social support may have the potential to augment the coping skills of patients with COPD by strengthening their ability to solve problems, stimulating greater motivation to take appropriate actions [103], and promoting their adherence to medication [24]. Consequently, our findings suggest that higher levels of social support could have a beneficial effect on mental health, particularly in patients with COPD, by helping them to better cope with their symptoms and aspects of the disease [24,103], and thus reducing the symptoms of depression and anxiety and improving their overall psychological well-being [82,104,105,106,107,108,109].

Research on the association between social support and sleep disturbances in patients with COPD is lacking, despite the extensive research conducted on the prevalence of poor sleep quality in these patients [9,14,110]. Sleep disturbances are prevalent in these patients and have been found to be associated with unfavorable clinical outcomes and reduced quality of life [14]. The results of our study indicated a high prevalence of poor sleep quality (69%) and symptoms of insomnia (75.5%), mirroring findings from prior studies [9,13,111,112,113,114]. Our study also revealed a higher prevalence of poor sleep quality and insomnia symptoms in patients with lower social support. The presence of low social support was strongly associated with insomnia symptoms, with a significant odds ratio of 5.17, even after accounting for potential confounders. However, this was not the case for sleep quality, although there was a borderline association between PSQI and MSPSS scores. Unfortunately, there is a scarcity of relevant studies supporting these results, with only one previous study demonstrating an association between being married and poor sleep quality, as assessed by PSQI [115]. Nevertheless, considering the potential importance of sleep patterns in managing COPD and overall well-being [116], there is a need for more targeted research on how social support affects sleep quality and insomnia symptoms in individuals with COPD. Conversely, our study also found a low prevalence of sleepiness in our sample, which is consistent with previous research [117,118,119]. In addition, we did not find any association between social support and daytime sleepiness; however, this could be explained by the small number of patients who reported sleepiness. Moreover, these findings suggest the need for further research to determine the role of social support in daytime sleepiness among these patients.

Our findings could help improve the management of patients with COPD and the development of interventions for these patients. Based on our analyses, it is evident that insufficient social support is associated with worse health outcomes, as explored using various PROMs. This could have a significant impact on the overall quality of life and well-being of patients with COPD. Especially in primary care settings where healthcare professionals have greater face-to-face interaction with patients, they have the opportunity to utilize PROMs and implement interventions that improve social support.

To the best of our knowledge, this is the first study in Greece that examines the link between social support and different PROMs, including sleep health. Moreover, to date, this study is the first to simultaneously assess all these PROMs and their relationship with the perceived social support of patients with COPD in primary care. However, our study had a few limitations. First, we utilized a cross-sectional research approach, which prevented us from establishing causal relationships between social support and PROMs. Despite the limitations in providing conclusive evidence, cross-sectional studies provide important insights into the relationships between different variables and could aid in the design of future prospective studies. Therefore, future research should employ a longitudinal approach that includes additional important measures, such as mortality rates and hospitalizations resulting from exacerbations, and overall quality of life. Second, the findings cannot be generalized due to the limited study sample, which consisted of only patients with COPD from six primary healthcare centers in Southern Greece, Crete. Third, it is difficult to compare our results with other studies that have used different methods to assess social support. Finally, the majority of the patients included in our study fell into categories A and B of the GOLD classification, while only a small number belonged to group E. This limited the applicability of our findings to populations with more severe COPD populations.

## 5. Conclusions

In conclusion, lower levels of social support were positively linked with several PROMs, such as worse health status, increased fatigue, depression, anxiety, and insomnia symptoms. Therefore, lower levels or a lack of social support could contribute to lower overall health among these patients. Policymakers and healthcare professionals could utilize our findings by implementing the evaluation of social support and not only the various PROMs to everyday clinical practice. Implementing this holistic approach to evaluate and improve COPD management has the potential to enhance the overall health of patients, especially in primary care settings.

## Figures and Tables

**Table 1 healthcare-12-00544-t001:** Demographic characteristics of the participants (n = 168) according to social support status.

Characteristics	Overall	High Support MSPPS > 5	Low Support MSPPS ≤ 5	*p*-Value
	N = 168 (100%)	N = 54 (32.1%)	N = 114 (68.9%)	
**Age (years)**	68.4 ± 9.0	66.3 ± 8.1	69.4 ± 9.3	0.040
Age group 40–50 years	8 (4.8)	3 (5.6)	5 (4.4)	0.175
Age group 51–64 years	53 (31.5)	22 (40.7)	31 (27.2)	
Age group ≥ 65 years	107 (63.7)	29 (53.7)	78 (68.4)	
**Gender**				
Male	114 (67.9)	39 (72.2)	75 (65.8)	0.404
Female	54 (32.1)	15 (27.8)	39 (34.2)	
**BMI**	29.8 ±6.1	29.6 ±4.9	29.8 ±6.7	0.841
**BMI ≥ 30**	69 (41.1)	20 (37.0)	49 (43.0)	0.464
**Have a spouse**				
Yes	40 (23.8)	8 (14.8)	32 (28.1)	0.060
No	128 (76.2)	46 (85.2)	82 (71.9)	
**Smoking**				
Active	76 (45.5)	21 (38.9)	55 (48.7)	0.395
Former	74 (44.3)	28 (51.9)	46 (40.7)	
Never	17 (10.2)	5 (9.3)	12 (10.6)	
Pack Years	65.1 ±37.2	60.7 ±39.4	67.3 ±36.1	0.309
**Alcohol (units/week)**	0.0 (0.0, 7.0)	2.0 (0.0, 10.0)	0.0 (0.0, 7.0)	0.137
**Physical Exercise (min/week)**	0.0 (0.0, 210.0)	0.0 (0.0, 210.0)	0.0 (0.0, 200.0)	0.276
**Education level**				
Primary level	78 (47.9)	22 (41.5)	56 (50.9)	0.009
Secondary level	57 (35.0)	15 (28.3)	42 (38.2)	
Higher level	28 (17.2)	16 (30.2)	12 (10.9)	
**Comorbidities**				
Asthma	29 (17.3)	20 (17.5)	9 (16.7)	0.888
Arterial Hypertension	89 (53.0)	25 (46.3)	64 (56.1)	0.233
CVD	54 (32.1)	21 (38.9)	33 (28.9)	0.198
Diabetes type 2	49 (29.2)	18 (33.3)	31 (27.2)	0.413
Hyperlipidemia	59 (51.8)	33 (61.1)	92 (54.8)	0.255
Obstructive Sleep Apnea	24 (14.3)	10 (18.5)	14 (12.3)	0.281
Osteoporosis	21 (12.5)	2 (3.7)	19 (16.7)	0.018
Cancer	21 (12.5)	7 (13.0)	14 (12.3)	0.901
Depression	22 (13.1)	6 (11.1)	16 (14.0)	0.600
Anxiety Disorder	8 (4.8)	1 (1.9)	7 (6.1)	0.223
Comorbidities ≥ 1	149 (88.7)	48 (88.9)	101 (88.6)	0.955

Data are presented as N (%) for categorical variables and mean values  ±  SD or median (25th–75th percentile) for continuous variables; BMI: Body Mass Index; CVD: cardiovascular diseases.

**Table 2 healthcare-12-00544-t002:** Patient disease characteristics (n = 168).

Characteristics	Overall	High Support MSPPS > 5	Low Support MSPPS ≤ 5	*p*-Value
	N = 168 (100%)	N = 54 (32.1%)	N = 114 (68.9%)	
**CAT score**	12.1 ±5.7	9.2 ±5.1	13.4 ±5.4	<0.001
CAT score ≥ 10	112 (66.7)	24 (44.4)	88 (77.2)	<0.001
mMRC *				0.001
0	**28 (16.8)**	**18 (15.8)**	**10 (18.9)**	
1	**101 (60.5)**	**61 (53.5)**	**40 (75.5)**	
2	**37 (22.2)**	**34 (29.8)**	**3 (5.7)**	
3	**1 (0.6)**	**1 (0.9)**	**0 (0.0)**	
**Exacerbations in the past year, n(%)**				
≤1	143 (85.1)	49 (90.7)	94 (82.5)	0.159
≥2	25 (14.9)	5 (9.3)	20 (17.5)	
≥1 hospitalization	8 (4.8)	2 (3.7)	6 (5.3)	0.658
**GOLD groups (n%)**				
**A**	52 (31)	27 (50)	25 (21.9)	
**B**	88 (52.4)	22 (40.7)	66 (57.9)	<0.001
**E**	28 (16.7)	5 (9.3)	23 (20.2)	

Data are presented as mean values  ±  SD or median (25th–75th percentile), unless otherwise indicated; CAT: COPD Assessment Test; GOLD: Global Initiative for Chronic Obstructive Lung Disease (GOLD 2023).* None of the patients had an mMRC score of 4.

**Table 3 healthcare-12-00544-t003:** Descriptive statistics of the MSPSS questionnaire.

	Mean ± SD	High Support MSPPS > 5 N = 54 (32.1%)	Low Support MSPPS ≤ 5 N = 114 (68.9%)	Min–Max	25th–75th Percentile
MSPSS “significant other”	4.6 (0.9)	76 (45.2)	92 (54.8)	1–7	4.3–5
MSPSS “family”	4.7 (0.9)	95 (56.5)	73 (43.5)	1–7	4.3–5
MSPSS “friends”	4.0 (1.3)	59 (35.1)	109 (64.9)	1–7	3–5
MSPSS total	4.4 (1.0)	54 (32.1)	114 (67.9)	1–7	3.8–5

**Table 4 healthcare-12-00544-t004:** Questionnaire (PROMs) scores of the 168 patients according to their social support status.

Symptoms	Overall	High Support MSPPS > 5	Low Support MSPPS ≤ 5	*p*-Value
	N = 168 (100%)	N = 54 (32.1%)	N = 114 (68.9%)	
**Daytime symptoms**				
**Fatigue**				
FSS	39.5 (27.0, 48.0)	32.0 (18.0, 43.0)	45.0 (27.0, 54.0)	<0.001
FSS ≥ 36	107 (64.5)	26 (49.1)	81 (71.7)	0.005
**Daytime sleepiness**				
ESS	3.0 (3.0, 9.0)	3.0 (1.5, 8.0)	3.0 (3.0, 9.0)	0.081
ESS ≥ 11	26 (15.8)	7 (13.5)	19 (16.8)	0.583
**Depressive symptoms**				
PHQ-9	4.0 (2.5, 8.0)	3.0 (1.0, 5.0)	5.0 (3.0, 8.0)	<0.001
PHQ-9 ≥ 10	26 (15.5)	4 (7.4)	22 (19.3)	0.047
**Anxiety symptoms**				
GAD-7	7.0 (3.3, 8.0)	5.0 (3.0, 7.0)	7.0 (4.0, 9.0)	0.005
GAD-7 ≥ 10	29 (17.3)	5 (9.3)	24 (21.1)	0.059
**Nighttime symptoms**				
PSQI	7.2 (2.9)	6.1 (2.8)	7.6 (2.8)	0.017
PSQI > 5	80 (69.0)	15 (51.7)	65 (74.7)	0.020
**Insomnia symptoms**				
Athens Insomnia Scale Score	8.2 (4.1)	6.2 (4.3)	9.2 (3.6)	<0.001
Athens Insomnia Scale Score ≥ 6	123 (75.5)	29 (54.7)	94 (85.5)	<0.001

**Table 5 healthcare-12-00544-t005:** Adjusted associations between perceived social support (continuous scales) and PROMs, estimated by linear regression models.

Symptoms	N ^a^	MSPSS “Significant Other” (Range 1–7)		MSPSS “Family” (Range 1–7)		MSPSS “Friends” (Range 1–7)		MSPSS Total (Range 1–7)	
		Beta (95%CI)	*p*-Value	Beta (95%CI)	*p*-Value	Beta (95%CI)	*p*-Value	Beta (95%CI)	*p*-Value
**CAT score**	163	−0.03 (−0.06, −0.01)	0.01	−0.03 (−0.05, −0.01)	0.01	−0.03 (−0.06, 0.01)	0.10	−0.03 (−0.06, −0.01)	0.02
CAT score ≥ 10	163	−0.33 (−0.63, −0.03)	0.03	−0.32 (−0.61, −0.02)	0.04	−0.23 (−0.65, 0.2)	0.29	−0.29 (−0.6, 0.01)	0.06
mMRC	162								
1 vs. 0		0.14 (−0.26, 0.53)	0.496	0.07 (−0.32, 0.46)	0.738	0.18 (−0.36, 0.73)	0.507	0.13 (−0.27, 0.53)	0.526
2 or 3 vs. 0		−0.20 (−0.66, 0.26)	0.397	−0.11 (−0.56, 0.35)	0.644	−0.18 (−0.82, 0.47)	0.590	−0.16 (−0.63, 0.31)	0.501
**Daytime symptoms**									
**Fatigue**									
FSS (range 9–63)	161	−0.01 (−0.02, 0,00)	0.02	−0.01 (−0.02, 0,00)	0.02	−0.02 (−0.03, 0.00)	0.03	−0.01 (−0.02, 0,00)	0.01
FSS ≥ 36	161	−0.12 (−0.41, 0.18)	0.43	−0.19 (−0.48, 0.11)	0.21	−0.11 (−0.52, 0.30)	0.60	−0.14 (−0.44, 0.16)	0.37
**Daytime sleepiness**									
ESS (range 0–24)	160	−0.02 (−0.05, 0.01)	0.24	−0.02 (−0.05, 0.01)	0.19	−0.04 (−0.08, 0.01)	0.1	−0.03 (−0.06, 0.01)	0.12
ESS ≥ 11	160	−0.17 (−0.58, 0.24)	0.41	−0.23 (−0.63, 0.17)	0.26	−0.36 (−0.92, 0.21)	0.22	−0.25 (−0.67, 0.16)	0.23
**Depressive symptoms**									
PHQ-9 (range 0–27)	163	−0.05 (−0.08, −0.02)	<0.01	−0.06 (−0.09, −0.03)	<0.01	−0.07 (−0.12, −0.02)	<0.01	−0.06 (−0.09, −0.03)	<0.01
PHQ-9 ≥ 10	163	−0.52 (−0.9, −0.15)	0.01	−0.78 (−1.13, −0.42)	<0.01	−0.67 (−1.19, −0.15)	0.01	−0.66 (−1.03, −0.28)	<0.01
**Anxiety symptoms**									
GAD-7 (range 0–21)	163	−0.05 (−0.09, −0.02)	<0.01	−0.06 (−0.09, −0.02)	<0.01	−0.07 (−0.12, −0.02)	0.01	−0.06 (−0.1, −0.02)	<0.01
GAD-7 ≥ 10	163	−0.44 (−0.83, −0.05)	0.03	−0.4 (−0.78, −0.02)	0.04	−0.6 (−1.14, −0.06)	0.03	−0.48 (−0.87, −0.09)	0.02
**Nighttime symptoms**									
PSQI (range 0–21)	112	−0.01 (−0.07, 0.05)	0.78	−0.03 (−0.09, 0.04)	0.37	−0.02 (−0.11, 0.06)	0.59	−0.02 (−0.09, 0.04)	0.52
PSQI > 5	112	−0.09 (−0.5, 0.32)	0.67	−0.17 (−0.59, 0.25)	0.42	0.04 (−0.51, 0.59)	0.88	−0.07 (−0.48, 0.34)	0.73
**Insomnia symptoms**									
Athens Insomnia Scale Score (range 0–24)	158	−0.05 (−0.08, −0.01)	0.01	−0.05 (−0.08, −0.01)	0.01	−0.07 (−0.11, −0.02)	0.01	−0.05 (−0.09, −0.02)	<0.01
Athens Insomnia Scale Score ≥ 6	158	−0.53 (−0.85, −0.2)	<0.01	−0.47 (−0.8, −0.15)	<0.01	−0.74 (−1.2, −0.29)	<0.01	−0.58 (−0.91, −0.25)	<0.01

Effect estimates are expressed for a 1-unit increase in each of the continuous scales. All models were adjusted for the participants’ age, sex, education, and marital status. ^a^ N represents the number of participants that had available PROM data and were included in each model.

**Table 6 healthcare-12-00544-t006:** Adjusted associations between perceived low social support (binary variable, high support was set as the referent category) and PROMs, estimated by logistic regression models.

Symptoms	N ^a^	MSPSS “Significant Other” ≤ 5 N = 92 (54.8%)		MSPSS “Family” ≤ 5 N = 73 (43.5%)		MSPSS “Friends” ≤ 5 N = 109 (64.9%)		MSPSS Total ≤ 5 N = 114 (67.9%)	
		OR (95%CI)	*p*-Value	OR (95%CI)	*p*-Value	OR (95%CI)	*p*-Value	OR (95%CI)	*p*-Value
**CAT score**	163	1.15 (1.07, 1.23)	<0.01	1.08 (1.02, 1.15)	<0.01	1.11 (1.04, 1.2)	0.01	1.17 (1.08, 1.27)	<0.01
CAT score ≥ 10	163	3.88 (1.86, 8.08)	<0.01	3.44 (1.55, 7.61)	<0.01	2.96 (1.4, 6.26)	0.01	3.97 (1.86, 8.44)	<0.01
mMRC	162								
1 vs. 0		1.05 (0.43, 2.59)	0.912	0.70 (0.27, 1.77)	0.449	0.80 (0.31, 2.11)		0.80 (0.31, 2.09)	0.651
2 or 3 vs. 0		2.53 (0.85, 7.54)	0.095	0.96 (0.32, 2.82)	0.935	2.58 (0.73, 9.09)	0.141	5.61 (1.28, 24.6)	0.022
**Daytime symptoms**									
**Fatigue**									
FSS (range 9–63)	161	1.04 (1.01, 1.06)	0.01	1.04 (1.01, 1.07)	0.01	1.06 (1.03, 1.09)	<0.01	1.06 (1.03, 1.09)	<0.01
FSS ≥ 36	161	1.66 (0.84, 3.27)	0.15	2.05 (0.98, 4.26)	0.06	2.4 (1.15, 5.02)	0.02	2.74 (1.31, 5.74)	0.01
**Daytime sleepiness**									
ESS (range 0–24)	160	1.07 (0.99, 1.16)	0.07	1.11 (1.02, 1.2)	0.01	1.12 (1.02, 1.22)	0.01	1.08 (0.99, 1.17)	0.08
ESS ≥ 11	160	1.71 (0.65, 4.46)	0.28	2.68 (1.01, 7.12)	0.05	2.57 (0.82, 8.09)	0.11	1.94 (0.64, 5.84)	0.24
**Depressive symptoms**									
PHQ-9 (range 0–27)	163	1.11 (1.02, 1.21)	0.02	1.15 (1.05, 1.26)	<0.01	1.17 (1.05, 1.3)	<0.01	1.19 (1.07, 1.34)	<0.01
PHQ-9 ≥ 10	163	1.64 (0.66, 4.1)	0.29	3.00 (1.18, 7.6)	0.02	1.92 (0.66, 5.61)	0.23	3.1 (0.93, 10.36)	0.07
**Anxiety symptoms**									
GAD-7 (range 0–21)	163	1.1 (1.01, 1.21)	0.03	1.13 (1.03, 1.23)	0.01	1.13 (1.03, 1.25)	0.01	1.12 (1.02, 1.23)	0.02
GAD-7 ≥ 10	163	1.7 (0.67, 4.34)	0.26	2.5 (0.99, 6.31)	0.05	3.09 (0.96, 9.94)	0.06	2.57 (0.82, 8.12)	0.11
**Nighttime symptoms**									
PSQI (range 0–21)	112	1.08 (0.93, 1.25)	0.31	1.1 (0.95, 1.27)	0.21	1.2 (1, 1.45)	0.05	1.19 (1, 1.43)	0.06
PSQI > 5	112	1.42 (0.58, 3.51)	0.45	1.18 (0.47, 2.92)	0.73	1.94 (0.71, 5.26)	0.19	2.09 (0.78, 5.64)	0.14
**Insomnia symptoms**									
Athens Insomnia Scale Score (range 0–24)	158	1.18 (1.07, 1.29)	<0.01	1.14 (1.04, 1.25)	<0.01	1.21 (1.09, 1.34)	<0.01	1.22 (1.1, 1.36)	<0.01
Athens Insomnia Scale Score ≥ 6	158	4.72 (2.04, 10.93)	<0.01	5.39 (2, 14.51)	<0.01	4.81 (2.06, 11.25)	<0.01	5.17 (2.23, 12.01)	<0.01

Effect estimates are expressed for a 1-unit increase in each of the continuous scales. All models were adjusted for the participants’ age, sex, education, and marital status. ^a^ N represents the number of participants that had available PROM data and were included in each model.

## Data Availability

The data presented in this study are available on request from the corresponding author. The data are not publicly available due to privacy restrictions.

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
