# Peer review of "Exploring the Link between Social Support and Patient-Reported Outcomes in Chronic Obstructive Pulmonary Disease Patients: A Cross-Sectional Study in Primary Care"

_healthcare, 2024, doi:10.3390/healthcare12050544_

Round 1

Reviewer 1 Report

Comments and Suggestions for Authors

An interesting study that highlights the relationship between social support and symptoms in COPD patients.

A main point. In the discussion, the authors highlight the limitations of their study characterized by a cross-sectional design and based on a limited number of patients from a small area of ​​their country. However, it would seem appropriate for the authors to discuss another limitation of their study. As far as I can understand, the study makes no reference to any spirometric confirmation of the diagnosis of COPD, which is instead based on the diagnosis known by the general practitioner. In general practice, the high frequency of COPD diagnoses based on symptoms and not confirmed by a spirometry test is known, as is the high frequency of COPD conditions confirmed by spirometry and unknown to the general practitioner. In the absence of spirometric diagnoses, it would be appropriate for the authors to discuss what limitations to the conclusions of their study may arise from a disease diagnosis based predominantly on symptoms and harmful exposures and in a minority of cases on spirometric confirmation. A sensitivity analysis could be useful if it were conducted on the subgroup for which, I hope, spirometric confirmation of the disease is available.

A minor point: The description of the questionnaires used to measure social support and patient-reported outcomes is well done but very long. Generally, the bibliographical references to which the reader is referred are used and the cut-off values ​​for the categorized variables and the references for choosing these values ​​are indicated.

Author Response

Reviewer 1 Comments to Author:

Comments and Suggestions for Authors

An interesting study that highlights the relationship between social support and symptoms in COPD patients.

We are grateful for your comments and the opportunity to revise and resubmit the manuscript.

A main point. In the discussion, the authors highlight the limitations of their study characterized by a cross-sectional design and based on a limited number of patients from a small area of ​​their country. However, it would seem appropriate for the authors to discuss another limitation of their study. As far as I can understand, the study makes no reference to any spirometric confirmation of the diagnosis of COPD, which is instead based on the diagnosis known by the general practitioner. In general practice, the high frequency of COPD diagnoses based on symptoms and not confirmed by a spirometry test is known, as is the high frequency of COPD conditions confirmed by spirometry and unknown to the general practitioner. In the absence of spirometric diagnoses, it would be appropriate for the authors to discuss what limitations to the conclusions of their study may arise from a disease diagnosis based predominantly on symptoms and harmful exposures and in a minority of cases on spirometric confirmation. A sensitivity analysis could be useful if it were conducted on the subgroup for which, I hope, spirometric confirmation of the disease is available.

Thank you for your comment. All participants had a physician-diagnosed COPD confirmed with spirometry. Care in Greece is fragmented so the records of spirometry could be in the hands of pulmonologists or primary care physicians based on whom the patient chose to go for assessment, so unfortunately, we don’t have access. However, in order to have a prescription of an inhaler it’s obligatory to have at least once a COPD diagnosis confirmed with spirometry. So, we do feel the diagnosis of COPD was sure.

Page2, Line 89-91 “To be eligible for inclusion, patients were required to meet the following criteria: (a) be 40 years of age or older and have a physician-diagnosed COPD confirmed with spirometry”

A minor point: The description of the questionnaires used to measure social support and patient-reported outcomes is well done but very long. Generally, the bibliographical references to which the reader is referred are used and the cut-off values ​​for the categorized variables and the references for choosing these values ​​are indicated.

Thank you for your comment. The description of questionnaires was shortened as suggested.

Reviewer 2 Report

Comments and Suggestions for Authors

Bouloukaki and colleagues evaluated the associations between social support and patient-reported outcomes in 168 COPD patients with COPG from six primary care centers in Crete, Greece using a cross-sectional study. Overall the manuscript is well-written and this is an interesting study. I have a few comments that I think could help strengthen the presentation of the methods and results.

  • In Table 5, I did not see a significant difference in terms of the associations of symptoms with different MSPSS subscales. Is it because these subscales were highly correlated with each other? Could you use the Pearson or the Spearman correlation test to assess the associations between these subscales? In addition, is there a way to show which subscale explains most of the variances in the total MSPSS?
  • Could you provide one or two sentences in the statistical analysis section to clarify when conducting regression analysis, whether you fit each PROM separately or you included PROMs in the same model?
  • In table 5 and table 6, could you explain what the "N" represents? For example, the "N" that corresponds to PSQI > 5 in table 5 is 112, does it mean that 112 out of 168 patients had PSQI > 5 or 112 patients had PSQI data? If that’s the latter, could you explain why 46 patients failed to provide PSQI? 

Author Response

Reviewer 2 Comments to Author:

Comments and Suggestions for Authors

Bouloukaki and colleagues evaluated the associations between social support and patient-reported outcomes in 168 COPD patients with COPD from six primary care centers in Crete, Greece using a cross-sectional study. Overall the manuscript is well-written and this is an interesting study. I have a few comments that I think could help strengthen the presentation of the methods and results.

We are grateful for your comments and the opportunity to revise and resubmit the manuscript.

In Table 5, I did not see a significant difference in terms of the associations of symptoms with different MSPSS subscales. Is it because these subscales were highly correlated with each other? Could you use the Pearson or the Spearman correlation test to assess the associations between these subscales? In addition, is there a way to show which subscale explains most of the variances in the total MSPSS?

Thank you for this truly insightful comment. Indeed, the subscales are highly correlated to each other and to the total scale and we speculate that this is the reason why the associations are apparently very similar. In fact, the highest correlation is between the significant other subscale and the family subscale (rho=0.887, p<0.001) and the lowest between the family and the friends’ subscale (rho=0.685, p<0.001). Also, the correlations with the total scale are strong and highly significant, ranging from 0.898 for the friends’ subscale and 0.921 for the significant other subscale (all p-values<0.001). This is also confirmed by regression analysis, where approximately 86% of the variance of the total social support scale was explained by the significant other and the family subscale while the friends’ subscale accounted for 80% of the variance. This information is now included in the results and you can find it below for your convenience.

Page 7-8, Line 283-290 “The three subscales are highly correlated to each other and to the total social support scale. The highest correlation was observed between the “significant other” subscale and the “family” subscale (rho=0.887, p<0.001) and the lowest between the “family” and the “friends” subscale (rho=0.685, p<0.001). Also, the correlations with the total scale were strong and highly significant, ranging from 0.898 for the “friends” subscale and 0.921 for the “significant other subscale” (all p-values<0.001). The “significant other” and the “family” subscales accounted for the 86% of the variance of the total social support scale this percentage for the”friends” subscale was 80% (data not shown).”

Page 5, Line 239-240 “Continuous scales were correlated using the Pearson’s correlation coefficient.”

Could you provide one or two sentences in the statistical analysis section to clarify when conducting regression analysis, whether you fit each PROM separately or you included PROMs in the same model?

We agree that in our initial version of the statistical analysis we didn’t make it clear that each PROM was fitted separately; a significant aspect of our analysis. Thank you for the opportunity to clarify this. Please find below the modified text of our statistical analysis

Page 5, Line 241-243 “In order to assess the associations between the MSPSS and the studied PROMs, we employed linear regression for the continuous MSPSS scales and logistic regression for the dichotomized MSPSS scales. Each PROM was then fitted into a separate model.”

In table 5 and table 6, could you explain what the "N" represents? For example, the "N" that corresponds to PSQI > 5 in table 5 is 112, does it mean that 112 out of 168 patients had PSQI > 5 or 112 patients had PSQI data? If that’s the latter, could you explain why 46 patients failed to provide PSQI?

Thank you for your comment. The N in this context represents the number of participants with available PROM data (for each particular PROM). Indeed 46 patients failed to complete all the questions in PSQI questionnaire, thus a total score could not be computed and were therefore subsequently excluded from the analysis. We have added a footnote in both Tables to make this clear for the reader.

Page 10, Line 334-335 and Page 11, Line 346-347“N represents the number of participants that had available PROM data and were included in each model”.

Reviewer 3 Report

Comments and Suggestions for Authors

Bouloukaki et.al., explored the link between social support and PROMs in COPD Patients. Overall, the manuscript is comprehensive and highlights the major findings with insufficient social support is associated with worse health outcomes with various PROMs. These kind of stands will help to improve social life of patients.

Authors should include dyspnea scale or mMRC grade to assess the symptom status of patients with COPD which will provide more weightage to the manuscript.

Authors can include history of Asthma in the Comorbidities.

Authors can also correlate the role BMI and COPD.

Patient study is very much required to understand the disease progression. Authors are managed to collect samples from six primary care centers and combined the data is appreciated.

The study providing potential importance of sleep patterns in managing COPD. * This study offers a clear picture of how social support impacts the quality of sleep and symptoms of insomnia in people with COPD. Additionally, the authors demonstrated how a lack or low degree of social support may be linked to a patient's overall worsening health.

Authors should provide IRB number in the methodology. Authors should include dyspnea scale or mMRC grade to assess the symptom status of patients with COPD which will provide more weightage to the manuscript. Authors can include history of Asthma in the Co-morbidities. Authors can also correlate the role BMI and COPD.

Authors should add dyspnea scale or mMRC scale and bifurcate based on the symptom status of the patients. Authors should correlate based on the different stages, which will be helpful in correlating daytime and nighttime symptoms.

Authors should add dyspnea scale or mMRC grade in the table 1. Authors should add history of Asthma in the comorbidities. Please provide normal range for CAT score in table 1.

Author Response

Reviewer 3 Comments to Author:

Comments and Suggestions for Authors

Bouloukaki et.al., explored the link between social support and PROMs in COPD Patients. Overall, the manuscript is comprehensive and highlights the major findings with insufficient social support is associated with worse health outcomes with various PROMs. These kind of stands will help to improve social life of patients.

We are grateful for your comments and the opportunity to revise and resubmit the manuscript.

Authors should include dyspnea scale or mMRC grade to assess the symptom status of patients with COPD which will provide more weightage to the manuscript.

Thank you for your comment. mMRC scores were added as suggested in Table 2. 

Page 7, Line 274 “The mMRC scores are also presented in Table 2.”

Authors can include history of Asthma in the Comorbidities.

Thank you for your comment. Prevalence of comorbid asthma was also added in Table 1.

Authors can also correlate the role BMI and COPD.

Thank you for your comment. BMI was not associated with COPD health status assessed by CAT score (either as continuous or dichotomous variable) in our study.

Patient study is very much required to understand the disease progression. Authors are managed to collect samples from six primary care centers and combined the data is appreciated. The study providing potential importance of sleep patterns in managing COPD. * This study offers a clear picture of how social support impacts the quality of sleep and symptoms of insomnia in people with COPD. Additionally, the authors demonstrated how a lack or low degree of social support may be linked to a patient’s overall worsening health.

Authors should provide IRB number in the methodology.

Thank you for your comments. IRB number was added in the methodology as suggested.

Page3, Line 102-104 “The study adhered to the guidelines specified in the Declaration of Helsinki and received approval from the University of Crete Research Ethics Committee (REC-UOC) (Protocol Number: 183/13.12.2022)”.

Authors should include dyspnea scale or mMRC grade to assess the symptom status of patients with COPD which will provide more weightage to the manuscript. Authors can include history of Asthma in the Co-morbidities. Authors can also correlate the role BMI and COPD.

Thank you for your comments. As stated above, mMRC scores were added as suggested in Table 2 and prevalence of comorbid asthma was also added in Table 1.

BMI was not associated with COPD health status assessed by CAT score (either as continuous or dichotomous variable) in our study.

Authors should add dyspnea scale or mMRC scale and bifurcate based on the symptom status of the patients. Authors should correlate based on the different stages, which will be helpful in correlating daytime and nighttime symptoms.

Thank you for your comments. We have added relevant data on Tables 5 and 6.

Authors should add dyspnea scale or mMRC grade in the table 1. Authors should add history of Asthma in the comorbidities.

As stated above, mMRC scores were added as suggested in Table 2 and prevalence of comorbid asthma was also added in Table 1.

Please provide normal range for CAT score in table 1.

Thank you for your comments. The interpretation of the CAT score is already provided in the methods section, page 3, Line 137-139 “The score ranges from 0 to 40, with higher values indicating poorer health status. A cutoff point of 10 or above is used to determine the presence of poor health status.”

Round 2

Reviewer 1 Report

Comments and Suggestions for Authors

No further comments.